# SU(3) hybrid static potentials at small quark-antiquark separations from fine lattices

Carolin Schlosser[1,2] and Marc Wagner[1,2*]

**1** Goethe-Universität Frankfurt am Main, Institut für Theoretische Physik,
Max-von-Laue-Straße 1, D-60438 Frankfurt am Main, Germany
**2** Helmholtz Research Academy Hesse for FAIR, Campus Riedberg,
Max-von-Laue-Straße 12, D-60438 Frankfurt am Main, Germany

⋆ mwagner@itp.uni-frankfurt.de

## Abstract

We summarize our recent lattice gauge theory computation of the $\Pi_u$ and $\Sigma_u^-$ hybrid static potentials at small quark-antiquark separations. We provide parameterizations of the resulting lattice data points, which can be used for investigating masses and properties of heavy hybrid mesons in the Born-Oppenheimer approximation.

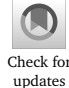
## 1 Introduction

The main goal of this work is to carry out a high precision first principles SU(3) lattice gauge theory computation of hybrid static potentials, i.e. potentials corresponding to a static quark antiquark pair and an excited gluonic flux tube with quantum numbers different from the ground state. Such potentials can e.g. be used to predict masses of $\bar{b}b$ and $\bar{c}c$ hybrid mesons within the Born-Oppenheimer approximation (for recent work discussing and using the Born-Oppenheimer approximation in the context of heavy hybrid mesons see Refs. [1–7]).

Hybrid static potentials have been computed with lattice gauge theory a number of times by independent groups [4, 8–34]. The majority of these computations were performed at a rather coarse lattice spacing. In this work we focus on the $\Pi_u$ and $\Sigma_u^-$ hybrid static potentials, which are the lowest hybrid static potentials. We consider four different lattice spacings as small as $a = 0.040\,\text{fm}$, which allows to identify and remove lattice discretization errors and also to study significantly smaller quark-antiquark separations $r$ than before. In particular our lattice results confirm the repulsive behavior of the $\Pi_u$ and $\Sigma_u^-$ hybrid static potentials at small $r$ predicted perturbatively in the framework of potential Non Relativistic QCD (pNRQCD) [2,35].

This contribution to the conference proceedings of the "XXXIII International (ONLINE) Workshop on High Energy Physics" summarizes our more detailed recent publication [36]. Results obtained at an early stage of this project have been published in Refs. [37, 38].

## 2 Hybrid static potential trial states and their quantum numbers

Hybrid static potentials can be characterized by the following quantum numbers:

- Absolute total angular momentum with respect to the quark-antiquark separation axis (e.g. the $z$ axis): $\Lambda = 0, 1, 2, \ldots \equiv \Sigma, \Pi, \Delta, \ldots$

- Parity combined with charge conjugation: $\eta = +, - = g, u$.

- Reflection along an axis perpendicular to the quark-antiquark separation axis (e.g. the $x$ axis): $\epsilon = +, -$.

For $\Lambda \geq 1$ static potentials are degenerate with respect to $\epsilon$. Thus, it is common to quote quantum numbers $\Lambda_\eta^\epsilon$ for $\Lambda = \Sigma$ and quantum numbers $\Lambda_\eta$ for $\Lambda = \Pi, \Delta, \ldots$ The ordinary static potential has quantum numbers $\Lambda_\eta^\epsilon = \Sigma_g^+$ and is denoted as $V_{\Sigma_g^+}(r)$. In this work we focus on the two lowest hybrid static potentials, which have quantum numbers $\Lambda_\eta^\epsilon = \Pi_u, \Sigma_u^-$ and are denoted as $V_{\Pi_u}(r)$ and $V_{\Sigma_u^-}(r)$.

To determine (hybrid) static potentials $V_{\Lambda_\eta^\epsilon}(r)$ using lattice gauge theory, one has to compute temporal correlation functions

$$W_{S,S';\Lambda_\eta^\epsilon}(r, t) = \langle \Psi_{\text{hybrid}}(t)|_{S;\Lambda_\eta^\epsilon} |\Psi_{\text{hybrid}}(0)\rangle_{S';\Lambda_\eta^\epsilon} \sim_{t\to\infty} \exp\left(-V_{\Lambda_\eta^\epsilon}(r)t\right) \tag{1}$$

of suitably designed trial states $|\Psi_{\text{hybrid}}\rangle_{S;\Lambda_\eta^\epsilon}$. From the asymptotic behavior for large $t$ one can extract $V_{\Lambda_\eta^\epsilon}(r)$. We use trial states

$$|\Psi_{\text{hybrid}}\rangle_{S;\Lambda_\eta^\epsilon} = \bar{Q}(-r/2) a_{S;\Lambda_\eta^\epsilon}(-r/2, +r/2) Q(+r/2)|\Omega\rangle \tag{2}$$

with static quark operators $\bar{Q}(-r/2)$ and $Q(+r/2)$ and gluonic parallel transporters

$$a_{S;\Lambda_\eta^\epsilon}(-r/2, +r/2) =$$

$$= \frac{1}{4} \sum_{k=0}^{3} \exp\left(\frac{i\pi\Lambda k}{2}\right) R\left(\frac{\pi k}{2}\right) \Big( U(-r/2, r_1)\Big(S(r_1, r_2) + \epsilon S_{\mathcal{P}_x}(r_1, r_2)\Big) U(r_2, +r/2) +$$

$$U(-r/2, -r_2)\Big(\eta S_{\mathcal{P}\circ\mathcal{C}}(-r_2, -r_1) + \eta\epsilon S_{(\mathcal{P}\circ\mathcal{C})\mathcal{P}_x}(-r_2, -r_1)\Big) U(-r_1, +r/2)\Big) \tag{3}$$

generating quantum numbers $\Lambda_\eta^\epsilon$ (for a detailed discussion we refer to Ref. [4]). On the lattice these gluonic parallel transporters are products of gauge links. To optimize $a_{S;\Lambda_\eta^\epsilon}(-r/2, +r/2)$, we have explored a large number of shapes and variations of their extents (see again Ref. [4]). For the computation of the $\Pi_u$ and $\Sigma_u^-$ hybrid static potentials we used those two operators with the largest ground state overlap ($S_{III,1}$ and $S_{IV,2}$ in Table 3 and Table 5 of Ref. [4]).

## 3 Lattice gauge theory computation of the ordinary static potential and the $\Pi_u$ and $\Sigma_u^-$ hybrid static potentials

We carried out computations of the ordinary (i.e. $\Sigma_g^+$) static potential and the $\Pi_u$ and $\Sigma_u^-$ hybrid static potentials on four ensembles (denoted as $A$, $B$, $C$ and $D$) with lattice spacings

Table 1: Gauge link ensembles used in this work (physical units are introduced by setting $r_0 = 0.5\,\text{fm}$).

| ensemble | $\beta$ | $a$ in fm | $(L/a)^3 \times T/a$ |
|----------|---------|-----------|----------------------|
| $A$ | 6.000 | 0.093 | $12^3 \times 26$ |
| $B$ | 6.284 | 0.060 | $20^3 \times 40$ |
| $C$ | 6.451 | 0.048 | $26^3 \times 50$ |
| $D$ | 6.594 | 0.040 | $30^3 \times 60$ |
| $A^{\text{HYP2}}$ | 6.000 | 0.093 | $24^3 \times 48$ |

$a$ ranging from $a = 0.093\,\text{fm}$ down to $0.040\,\text{fm}$ (see Table 1). We used unsmeared temporal links, i.e. the standard Eichten-Hill static action, and APE smeared spatial links to maximize the ground state overlaps of the trial states discussed in the previous section. To reduce statistical errors, we employed a multilevel algorithm [39]. Moreover, we reuse the lattice data from our previous work [4] obtained at lattice spacing $a = 0.093\,\text{fm}$ with the HYP2 static action (the corresponding ensemble is denoted as $A^{\text{HYP2}}$).

In that way we get a fine spatial resolution of the potentials. Because of the rather small lattice spacings of ensemble $C$ and ensemble $D$, we are also able to access significantly smaller quark-antiquark separations than before (in lattice gauge theory one should only use lattice data points with $r \gtrsim 2\,a$, to avoid sizable discretization errors). Moreover, using five ensembles we are able to quantify and eliminate discretization errors.

We note that static potentials computed via correlation functions (1) have self energies, which depend both on the lattice spacing and the static quark action and diverge in the limit $a \to 0$. These self energies need to be subtracted, before all our lattice data points can be shown together in a meaningful plot. This is done by suitable fits and discussed in section 4.

We investigated and excluded the following types of systematic errors:

- **Errors due to topological freezing:**
  Since Monte Carlo algorithms have difficulties changing the topological charge $Q$ for lattice spacings $a \lesssim 0.05\,\text{fm}$ [40], Monte Carlo histories of $Q$ need to be checked, in particular for ensembles $C$ and $D$. We found that autocorrelation times of $Q$ are quite large for these two ensembles. We carried out very long simulations to guarantee that there is a sufficiently large number of changes in $Q$ such that the ensembles form representative sets of gauge link configurations distributed according to $e^{-S}$.

- **Finite volume corrections:**
  A finite spatial volume leads to a negative energy shift, because of virtual glueballs traveling around the far side of the periodic volume [41]. For very small volumes one expects positive energy shifts, because of squeezed wave functions [42], in particular for hybrid static potentials, where the flux tubes are quite extended [33, 43]. We studied the volume dependence of the $\Sigma_g^+$, $\Pi_u$ and $\Sigma_u^-$ static potentials in detail and found that both types of effects are negligible for spatial extent $L \gtrsim 1.2\,\text{fm}$, a condition fulfilled for all five ensembles we used (see Table 1).

- **Glueball decays:**
  At small $r$ hybrid flux tubes can decay into $\Sigma_g^+$ flux tubes and glueballs. In Ref. [36] we showed analytically that the $\Sigma_u^-$ flux tube is protected by symmetries from decays into a $0^{++}$ glueball. For the $\Pi_u$ flux tube decays into a $0^{++}$ glueball are possible for $r \lesssim 0.11\,\text{fm}$.

Numerically, however, we observed no indication that $V^e_{\Pi_u}(r)$ is contaminated by such decays. Since the $\Pi_u$ and $\Sigma^-_u$ potentials approach each other for small $r$, glueball decays seem to have a negligible effect on $V^e_{\Pi_u}(r)$.

For a more detailed discussion on the exclusion of systematic errors we refer to our recent publication [36].

# 4 Parameterization of the ordinary static potential and the $\Pi_u$ and $\Sigma^-_u$ hybrid static potentials

In a preparatory step we determined a parameterization $V_{\Sigma^+_g}(r)$ of the lattice data points for the ordinary static potential. This is important, because we obtained the ensemble dependent self energies rather precisely and we were able to estimate lattice discretization errors at tree-level of perturbation theory. This information will be used below to determine parameterizations for the $\Pi_u$ and $\Sigma^-_u$ hybrid static potentials. Moreover, $V_{\Sigma^+_g}(r)$ is useful to set the energy scale, when interpreting the static quarks as either $b$ quarks or $c$ quarks. To do this one can compute the quarkonium ground state $\eta_b(1S) \equiv \Upsilon(1S)$ or $\eta_c(1S) \equiv J/\Psi(1S)$ in the Born-Oppenheimer approximation and identify the result with the corresponding experimental result.

We carried out an 8-parameter fit of the ansatz

$$V^{\text{fit},e}_{\Sigma^+_g}(r) = V_{\Sigma^+_g}(r) + C^e + \Delta V^{\text{lat},e}_{\Sigma^+_g}(r) \tag{4}$$

$$V_{\Sigma^+_g}(r) = -\frac{\alpha}{r} + \sigma r \tag{5}$$

$$\Delta V^{\text{lat},e}_{\Sigma^+_g}(r) = \alpha'\left(\frac{1}{r} - \frac{G^e(r/a)}{a}\right) \tag{6}$$

to the $\Sigma^+_g$ data points from all five ensembles with $r \geq 0.2\,\text{fm}$. $V_{\Sigma^+_g}(r)$ is the Cornell ansatz, which provides an accurate description of the ordinary static potential for $r \gtrsim 0.2\,\text{fm}$ (see e.g. Ref. [44]). $C^e$ denote the $a$-dependent self energies. $G^e(r/a)/a$ is proportional to the ordinary static potential at tree-level of lattice perturbation theory, i.e. it is the lattice counterpart of $1/r$ in the continuum. Thus, $\Delta V^{\text{lat},e}_{\Sigma^+_g}(r)$ represent lattice discretization errors at tree-level of perturbation theory.

The resulting fit parameters allow to define data points, with the self-energy subtracted and discretization errors removed,

$$\tilde{V}^e_{\Sigma^+_g}(r) = V^e_{\Sigma^+_g}(r) - C^e - \Delta V^{\text{lat},e}_{\Sigma^+_g}(r). \tag{7}$$

These improved lattice data points together with the parameterization (4) are shown in Figure 1.

Similarly, we carried out a 10-parameter fit

$$V^{\text{fit},e}_{\Lambda^\epsilon_\eta}(r) = V_{\Lambda^\epsilon_\eta}(r) + C^e + \Delta V^{\text{lat},e}_{\text{hybrid}}(r) + A'^e_{2,\Lambda^\epsilon_\eta} a^2 \quad , \quad \Lambda^\epsilon_\eta = \Pi_u, \Sigma^-_u \tag{8}$$

$$V_{\Pi_u}(r) = \frac{A_1}{r} + A_2 + A_3 r^2 \quad , \quad V_{\Sigma^-_u}(r) = \frac{A_1}{r} + A_2 + A_3 r^2 + \frac{B_1 r^2}{1 + B_2 r + B_3 r^2} \tag{9}$$

$$\Delta V^{\text{lat},e}_{\text{hybrid}}(r) = -\frac{1}{8}\Delta V^{\text{lat},e}_{\Sigma^+_g}(r), \tag{10}$$

to the $\Pi_u$ and $\Sigma^-_u$ data points from all five ensembles with $r \geq 2a$. $V_{\Pi_u}(r)$ and $V_{\Sigma^-_u}(r)$ are parameterizations of the $\Pi_u$ and $\Sigma^-_u$ hybrid static potentials consistent with and motivated by

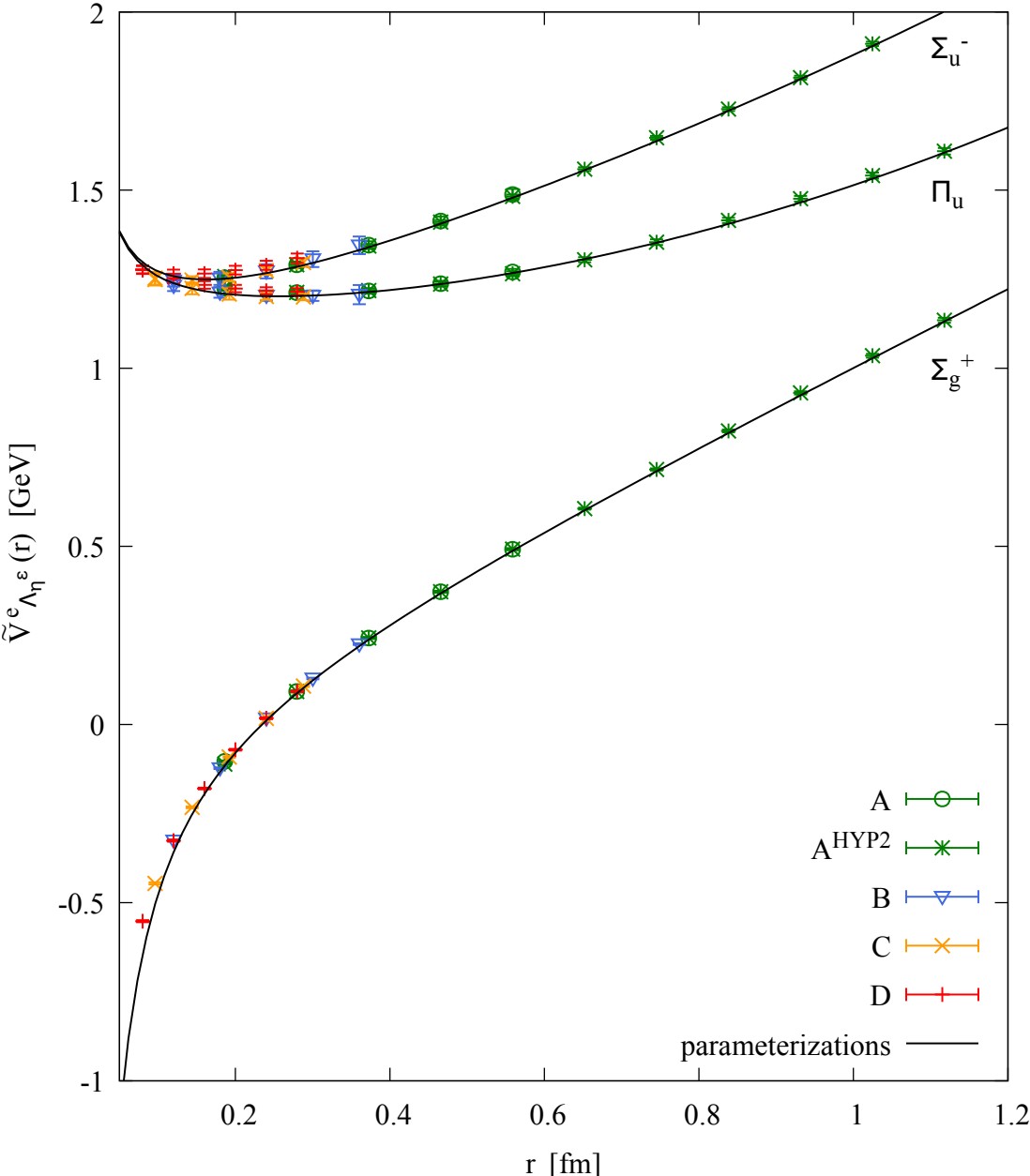

Figure 1: Improved lattice data points (7) and (11) together with the parameterizations (5) and (9) for the ordinary static potential and the $\Pi_u$ and $\Sigma_u^-$ hybrid static potentials.

the pNRQCD prediction at small $r$ [2, 35]. As before, $C^e$ denote the $a$-dependent self energies and $\Delta V^{\text{lat},e}_{\text{hybrid}}(r)$ lattice discretization errors at tree-level of perturbation theory. Moreover, $A'^e_{2,\Lambda^\epsilon_\eta} a^2$ represent the leading order lattice discretization errors in the difference to the ordinary static potential, which turned out to be sizable.

In analogy to Eq. (7), the resulting fit parameters allow to define data points, with the self-energy subtracted and discretization errors removed,

$$\tilde{V}^e_{\Lambda^\epsilon_\eta}(r) = V^e_{\Lambda^\epsilon_\eta}(r) - C^e - \Delta V^{\text{lat},e}_{\text{hybrid}}(r) - A'^e_{2,\Lambda^\epsilon_\eta} a^2. \tag{11}$$

These improved lattice data points together with the parameterizations (9) are shown in Figure 1.

## 5   Summary and conclusions

We used lattice gauge theory to compute the $\Pi_u$ and $\Sigma_u^-$ hybrid static potentials at four different lattice spacings, where the smallest lattice spacing $a = 0.040\,\text{fm}$ is significantly smaller than lattice spacings used in the majority of existing computations. This allows us to provide lattice data points for quark-antiquark separations as small as $0.08\,\text{fm}$. By carrying out suitable fits we subtracted the ensemble dependent self energies and removed lattice discretization errors to a large extent. Moreover, various systematic errors were checked and excluded.

The resulting parameterizations (5) and (9) differ from those obtained in our earlier work [4], where only one ensemble with rather coarse lattice spacing was available. A simple single channel Born-Oppenheimer prediction of heavy hybrid meson masses led to discrepancies between 10 MeV and 45 MeV (see Ref. [36]). Thus, it is expected that the high quality lattice data discussed in this work or, equivalently, the resulting parameterizations (5) and (9) will lead to a significant gain in precision, when used in recently developed more sophisticated Born-Oppenheimer approaches, which include coupled channels and heavy spin corrections [2, 3, 5, 6].

We note that the bare lattice data points, the improved lattice data points (7) and (11) and the parameterizations (5) and (9) are provided in detail in Ref. [36].

## Acknowledgements

M.W. thanks the organizers of the "XXXIII International (ONLINE) Workshop on High Energy Physics" for the invitation and possibility to give a talk.

We thank Christian Reisinger for providing his multilevel code. We acknowledge interesting and useful discussions with Eric Braaten, Nora Brambilla, Francesco Knechtli, Colin Morningstar, Lasse Müller, Christian Reisinger and Joan Soto.

M.W. acknowledges support by the Heisenberg Programme of the Deutsche Forschungsgemeinschaft (DFG, German Research Foundation) – project number 399217702.

Calculations on the GOETHE-HLR and on the on the FUCHS-CSC high-performance computers of the Frankfurt University were conducted for this research. We would like to thank HPC-Hessen, funded by the State Ministry of Higher Education, Research and the Arts, for programming advice.

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
