# Peer review of "SU(3) hybrid static potentials at small quark-antiquark separations from fine lattices"

_SciPost Physics Proceedings, doi:SciPost Phys. Proc. 6, 009 (2022)_

## Round 1 · Referee Report · Anonymous · 2022-3-29

Strengths

Systematic and important calculations in lattice phenomenology are performed, clear presentation

Weaknesses

No comments are made concerning the scope of validity of the static-potential approach to the description of hybrid quarkonia so it is not clear whether the corrections due to lattice artifacts are smaller or greater than those caused by deviations from Born-Oppenheimer approximation, relativistic effects etc.

Report

Currently theoretical studies of exotic mesons represent a very hot topic. The authors consider a particularly interesting sort of exotica - heavy hybrid mesons. They calculate hybrid static potentials with several quantum numbers (associated with the energy of excited gluon fields) using the ordinary static potential as a reference. Due regard for the lattice artifacts is provided. Sufficiently fine lattices are used making it possible to evaluate the hybrid potential down to 0.08 fm, which can be used as input for the phenomenological predictions for heavy hybrid quarkonia in the Born--Oppenheimer approximation.

  • validity: top
  • significance: high
  • originality: top
  • clarity: high
  • formatting: perfect
  • grammar: excellent

Author:  Marc Wagner  on 2022-03-29  [id 2337]

(in reply to Report 1 on 2022-03-29)
Category:
remark
answer to question

We thank the referee for his/her positive report.

No comments are made concerning the scope of validity of the static-potential approach to the description of hybrid quarkonia so it is not clear whether the corrections due to lattice artifacts are smaller or greater than those caused by deviations from Born-Oppenheimer approximation, relativistic effects etc.

This is certainly a very interesting question. We address the question briefly in section 5:

"The resulting parameterizations (5) and (9) differ from those obtained in our earlier work [4], where only one ensemble with rather coarse lattice spacing was available. A simple single channel Born-Oppenheimer prediction of heavy hybrid meson masses led to discrepancies between 10MeV and 45MeV (see Ref. [36]). Thus, it is expected that the high quality lattice data discussed in this work or, equivalently, the resulting parameterizations (5) and (9) will lead to a significant gain in precision, when used in recently developed more sophisticated Born-Oppenheimer approaches, which include coupled channels and heavy spin corrections [2,3,5,6]."

At the moment this is all we know and can say. We are, however, in the process of computing heavy spin corrections and implementing an up-to-date Born-Oppenheimer approach. Once we have results available, we plan to discuss the raised question in more detail in a future publication.

---

## Editorial Decision

published